# Remote Monitoring of Human Vital Signs Based on 77-GHz mm-Wave FMCW Radar

**DOI:** 10.3390/s20102999

**Published:** 2020-05-25

**Authors:** Yong Wang, Wen Wang, Mu Zhou, Aihu Ren, Zengshan Tian

**Affiliations:** School of Communication and Information Engineering, Chongqing University of Posts and Telecommunications, Chongqing 400065, China; wwangwen96@163.com (W.W.); zhoumu@cqupt.edu.cn (M.Z.); rah15320299632@163.com (A.R.); tianzs@cqupt.edu.cn (Z.T.)

**Keywords:** non-contact, frequency-modulated continuous waveform, orthogonal matching pursuit, discrete wavelet transform

## Abstract

In recent years, non-contact radar detection technology has been able to achieve long-term and long-range detection for the breathing and heartbeat signals. Compared with contact-based detection methods, it brings a more comfortable and a faster experience to the human body, and it has gradually received attention in the field of radar sensing. Therefore, this paper extends the application of millimeter-wave radar to the field of health care. The millimeter-wave radar first transmits the frequency-modulated continuous wave (FMCW) and collects the echo signals of the human body. Then, the phase information of the intermediate frequency (IF) signals including the breathing and heartbeat signals are extracted, and the Direct Current (DC) offset of the phase information is corrected using the circle center dynamic tracking algorithm. The extended differential and cross-multiply (DACM) is further applied for phase unwrapping. We propose two algorithms, namely the compressive sensing based on orthogonal matching pursuit (CS-OMP) algorithm and rigrsure adaptive soft threshold noise reduction based on discrete wavelet transform (RA-DWT) algorithm, to separate and reconstruct the breathing and heartbeat signals. Then, a frequency-domain fast Fourier transform and a time-domain autocorrelation estimation algorithm are proposed to calculate the respiratory and heartbeat rates. The proposed algorithms are compared with the contact-based detection ones. The results demonstrate that the proposed algorithms effectively suppress the noise and harmonic interference, and the accuracies of the proposed algorithms for both respiratory rate and heartbeat rate reach about 93%.

## 1. Introduction

Vital sign signals are important indicators in modern health care and medical applications [1]. Monitoring the vital parameters, such as heartbeat and breathing signals, provides doctors with reliable diagnosis and treatment basis. The traditional vital sign detection method mainly uses a contacted wearable sensor [2] or a sticky electrode [3] to directly monitor the heartbeat and breathing signals. However, both the wearable sensor-based and sticky electrode-based methods restrict the users’ behavior due to the wired lines, and thus they cannot truly reflect the change of the users’ vital characteristics parameters with unconscious movement. Moreover, the contact-based sensor is usually complicated to operate, and it often makes the users uncomfortable. In comparison, non-contact radar detection technology can monitor the heartbeat and breathing signals over long distances without electrodes or sensors, making it easier and more comfortable for users. More importantly, the non-contact radar sensor has broad applications in the field of rescue, counter-terrorism response, and emergency search [4]. As a result, non-contact radar detection technology has gradually become a research hotspot in the emerging application fields, such as vital sign monitoring [5], oceanographic surface current velocity measurements [6], naval search and rescue operations [7,8], position occupying perception [9,10], and gesture perception in human-computer interaction [11,12]. This paper focuses on the vital sign monitoring, and other application fields of non-contact radar sensing are out the scope of this paper.

At present, there are three main types of radar systems for vital sign monitoring, namely pulse radar [13] /(ultra-wide band radar) [14], continuous wave (CW) Doppler radar [15,16], and frequency modulated continuous wave (FMCW) radar [17]. In Reference [13], the authors reduce the noise through a time-varying filter to extract the heartbeat and respiration rates using a 3.3 GHz pulsed radar; however, the detection accuracy has not yet been discussed. Meanwhile, the pulse radar used in [13] is very unfavorable in the system integration and low-power operation due to its broad band characteristics. In contrast, CW and FMCW radars have higher levels of system integration and lower-power operation, and easier proximity detection. Thus, Dong et al. [16] apply a 24 GHz CW Doppler radar to obtain electrocardiograms, by detecting the combined atrial and ventricular contractions and relaxations conducted to the skin on the back. Since the CW Doppler radar lacks modulation spectrum information, it is not good at distinguishing clutter and multiple targets. As a result, the CW Doppler radar-based vital sign monitoring methods rely only on the Doppler information to detect the relative motion [18]. Moreover, since the Doppler radar gets only the doppler information and it cannot get the range information, the extracted doppler information may contain the micro-motion interference that from the human body itself and nearby environment, leading to a low vital sign detection accuracy. Compared to CW and pulse radars, the FMCW radar has a higher range and speed resolution, as well as its ability to distinguish multiple targets and extract target fretting information, making it a mainstream choice in the field of life detection. Therefore, this paper adopts FMCW radar for human non-contact vital sign detection.

The FMCW radar has been implemented for vital sign monitoring by Sharpe et al. in 1990 [19]. In 2013, Zhang et al. [20] use a 24.15 GHz FMCW radar with a scanning bandwidth of 72 MHz to extract the human heartbeat rate signal, and propose a projection matrix method for periodic clutter suppression. Although it considers the clutter interference of surrounding static objects, it does not eliminate the interference of human micro-motion on the heartbeat rate estimation. Since the micro-motion may cover the respiratory movement, the incorrect extracted heartbeat information may be incorrect. Lee et al. [21] use a 24 GHz FMCW radar for heartbeat rate estimation, and they eliminate the environmental clutter noise and reduce the mutual interference among multiple people. Nevertheless, the influence of micro-motion of human target on the characteristics estimation of living body is not considered. In [22], the authors apply a 9.6 GHz FMCW radar for heartbeat and respiration rate detection. To reduce the influence of human respiratory fretwork harmonics on the heartbeat rate estimation, Anitori et al. [22] propose two methods for heartbeat rate and respiratory rate calculation using the signal amplitude and phase information. However, the specific solutions for harmonic and noise reduction are inefficient. Gu et al. [23] use the arctangent demodulation and phase compensation for a 2.4 GHz radar to solve the problem of harmonic interference caused by body movement. Due to the use of orthogonal receivers, the I/Q signals introduces the DC offset, thus a more complex I/Q signal calibration is required. The high-frequency millimeter-wave radar provides a high signal-to-noise ratio, and has a high sensitivity when measuring the vital sign signals even with small displacements of tens of microns [24]. Alizadeh et al. [25], the authors apply a 77 GHz millimeter-wave radar to detect the components of vital signal by extracting the phase of the intermediate frequency (IF) signal. Such a 77 GHz millimeter-wave radar not only avoids the higher harmonics caused by the amplitude signal estimation, but also solves the problem of dc signal offset caused by the orthogonal receivers. Nevertheless, the method proposed in [25] may cause the phenomenon of respiratory harmonic frequencies that conceal the heartbeat frequency, leading to a decrease in the accuracy of heartbeat rate estimation. Alizadeh et al. [25] use two bandpass filters to separate the heartbeat and breathing signals and find the largest spectral values (in the two bands) as the estimated respiratory frequency and heartbeat frequency. However, because of the existed respiratory harmonic or noise, the obtained heartbeat and breathing frequencies might be incorrect, especially for the heartbeat one.

To address the aforementioned issues, this paper proposes to use a high-frequency millimeter-wave FMCW radar with a starting frequency of 77 GHz and a chirp frequency of 4 GHz to detect the human breathing and heartbeat signals. The main contributions are concluded as follows:(1)By briefly introducing the principle of FMCW radar, we induce the closed-form expression of the measurement accuracy of human heartbeat and respiration detection. In addition, the range determination of human body in the radar field of view and the vibration FFT of breathing and heartbeat are expounded.(2)The circular center dynamic tracking algorithm is applied to correct the DC offset. Then, the extended differential and cross-multiply (DACM) is applied for phase unwrapping to obtain correct phase change information of breathing and heartbeat signals, and the phase signal is further differentially enhanced.(3)We propose two methods, compressive sensing based on orthogonal matching pursuit (CS-OMP) and rigrsure adaptive soft threshold noise reduction based on discrete wavelet transform (RA-DWT), to separate and reconstruct the heartbeat and respiratory signals, and obtain interference-suppressed heartbeat and respiratory signals. Then, the two methods, namely frequency FFT algorithm and time domain autocorrelation algorithm, are presented to calculate the heartbeat and respiratory rates.(4)To verify the effectiveness of the proposed algorithms, the obtained heartbeat rate and respiration rate are respectively compared with the two contact-based sensors (smart bracelet Mi 3 and airflow sensors). The results show that the proposed CS-OMP and RA-DWT algorithms effectively suppress the harmonics and interference, and greatly improve the detection accuracy of heartbeat rate and respiratory rate.

The remainder of this paper is structured as follows: In Section 2, the mechanism of FMCW radar and the actual detection accuracy of the radar are discussed. In Section 3, detailed introduction to the extraction algorithm implementation steps are detailed. In Section 4, the results using the measured data are presented and discussed. Finally, conclusions are given in Section 5.

## 2. Theory of FMCW Radar

The FMCW radar system includes the radio frequency (RF) such as transmit (TX) and receive (RX), analog clocking, and digital circuit such as analog-to-digital converters (ADCs), microcontrollers (MCUs), and digital signal processors (DSPs). Figure 1 shows a simplified block diagram of a typical FMCW radar system. The FMCW radar sends a linear FMCW signal generated by the synthesizer. The radar signal will be reflected when it encounters an object. Then, the orthogonal receiver is responsible for capturing the echo signal and orthogonally mixing it with the transmitted signal. A low-pass filter is used to filter out the high frequency part and obtain the IF signal. Finally, the IF signal is sampled by an ADC.

The transmitted signals are usually sawtooth or triangle waves, and the sawtooth wave is used in this paper. It is noted that the method proposed in this paper is also suitable for the triangle wave, but more effort are required. The specific form is shown in Figure 2.

The transmitted signal of the FMCW radar can be expressed as [26]
(1)$$x_{T}\left(t\right)=A_{TX}\cos\left(2\pi f_{c}t+\pi \frac{B}{T_{c}}t^{2}+\theta \left(t\right)\right),$$
where *f_c_* is the starting frequency of the chirp signal, *B* is the bandwidth, *A_TX_* is the amplitude of the transmitted signal, θ(t) is the phase noise, *T_c_* is the width of chirp signal pulse, and *B/T*${}_{c}$ is the slope of the chirp signal, which represents the changing rate of the frequency.

Assuming that *R*(*t*) is the motion displacement of the chest, and *d${}_{0}$* is the distance from the radar sensor to the human body. The distance from the chest to the radar is *x*(*t*) = *R*(*t*) + *d*${}_{0}$, and the time delay is *t*${}_{d}$ = 2*x*(*t*)/*c*, where *c* is the light speed. Then, the received signal is
(2)$$x_{R}\left(t\right)=A_{RX}\left\{\cos(2\pi f_{c}\left(t-t_{d}\right))\right.+\pi \frac{B}{T_{c}}{\left(t-t_{d}\right)}^{2}+\theta \left(t-t_{d}\right)\left.\right\}.$$

The echo signal and the transmission signal are mixed by two orthogonal I/Q channels, and then passed through a low-pass filter to obtain an IF signal *S${}_{IF}$(t)*
(3)$$\begin{array}{ccc} S_{IF}\left(t\right) & = & A_{T}A_{R}\exp\left(j\left(2\pi \left[\frac{B}{T_{c}}t_{d}\right]t+2\pi f_{c}t_{d}+\pi \frac{B}{T_{c}}{t_{d}}^{2}+\Delta \theta \left(t\right)\right)\right)\\ & \approx & A_{T}A_{R}\exp\left(j\left(2\pi \left[\frac{B}{T_{c}}t_{d}\right]t+2\pi f_{c}t_{d}+\Delta \theta \left(t\right)\right)\right)\\ & \approx & A_{T}A_{R}\exp\left(j\left(2\pi \underset{f_{b}}{\underset{︸}{\frac{2Bd_{0}}{cT_{c}}}}t+\underset{\psi (t)}{\underset{︸}{\frac{4\pi (R\left(t\right)+d_{0})}{\lambda }}}\right)\right)\\ & = & A_{T}A_{R}\exp(j\left(2\pi f_{b}t+\psi \left(t\right)\right), \end{array}$$
where *t${}_{d}$* = $2(R\left(t\right)+d_{0})/c$. Since the third term, $\pi \frac{B}{T_{c}}{t_{d}}^{2}$, is very small (about $10^{6}$ order) in the practical scenario, it is ignored in the second approximation. The residual noise phase term $\Delta \theta \left(t\right)=\theta \left(t\right)-\theta \left(t-t_{d}\right)$ can also be neglected in the third approximation because of the range correlation effects [27]. In general, t=1μs , the chest displacement R(t) is in mm, and R(t)t can be ignored. It can be seen that at a fixed distance $d_{0}$, ψ(t) varies with R(t) relative to λ.

Since the velocity rate of the chest motion is much lower than $10^{3}$ m/s, the chest displacement is no more than 1 mm per chirp (duration time of a single chirp is 50 μs). Thus, R(t) will be very small and it approximates a constant in the microsecond time of a single chirp. Therefore, in order to obtain the change in chest displacement, multiple chirps should be transmitted in sequence to obtain the chest displacement information, which is equivalent to sampling R(t). Assuming that *L* chirp signals are sent, and the sampling interval for R(t) is the frame period $T_{m}$. When $T_{m}\geq T_{c}$, the chest displacement information related to the change of R(t) will be obtained at the distance of the human target. The delay will become
(4)$$t_{d}=\frac{2\left[d_{0}+R\left(lT_{c}+t\right)\right]}{c}.$$

The form of displacement movement caused by thoracic dilatation is the form of cosine signal, which satisfies $R\left(lT_{c}+t\right)\approx R\left(lT_{c}\right)$. Substituting Equation (Equation 4) into Equation (Equation 3), and, after sampling, we can obtain
(5)$$S_{IF}\left(lT_{c}+nT_{s}\right)=A_{T}A_{R}\exp(j\left(2\pi f_{b}nT_{s}+\psi_{l}\right),$$
(6)$$f_{b}=\frac{2Bd_{0}}{cT_{c}}\;\psi_{l}=\frac{4\pi (R\left(lT_{c}\right)+d_{0})}{\lambda },$$
where $T_{s}$ is the fast-time sampling period. The acquired beat frequency signal is a two-dimensional function of fast sampling *n* and slow sampling $lT_{c}$. Therefore, we should find the information of $f_{b}$ and $\psi_{l}$ from the sampled beat signal to estimate the distance and the vibration frequency caused by the thorax. These two pieces of information can be obtained using a FFT. It can be found that, when the FFT is performed, only the phase term $\psi_{l}$ is left if $f=f_{b}$. The FFT is performed on a single chirp on the fast sampling time axis to obtain the spectrum of the beat signal. The peak values of the spectrum correspond to the targets at different distances, which is called range FFT [28]. A FFT is performed on the slow time axis to obtain the vibration frequency, which is called vibration FFT [29].

The maximum detection range of FMCW radar is
(7)$$d_{\max}=\frac{F_{s}c}{2S},$$
where $F_{s}$ is the sampling rate. Substituting $S=\frac{B}{T_{c}}$ into Equation (Equation 7), we have
(8)$$d_{\max}=\frac{F_{s}T_{c}c}{2B}=M_{\mathrm{samples}}\frac{c}{2B}=M_{\mathrm{samples}}d_{res},$$
where $M_{\mathrm{samples}}=F_{s}T_{c}$ is the number of sampling points in the period of $T_{c}$. The theoretical range resolution of FMCW radar is defined by $d_{res}=c/2B$ according to the Nyquist sampling theorem. It can be seen that the effective ranging $d_{max}$ is proportional to the distance resolution $d_{res}$. Combining Equations (Equation 7) and (Equation 8), the range resolution can also describe as
(9)$$d_{real\_res}=\frac{1}{M_{\mathrm{samples}}}\frac{F_{s}T_{c}c}{2B}.$$

It should be noted that the range resolution may not reach the theoretical value due to the equipment in the actual experiment. Similarly, the vibration frequency caused by the chest cavity is related to the sampling frame rate R(t) of $1/T_{m}$. The Nyquist sampling principle limits the maximum vibration frequency, and the minimum vibration frequency is determined by the number of FFT chirps *N*, thereby giving the actual maximum and minimum frequencies
(10)$$f_{\max}=\frac{1}{2T_{m}}\;f_{\min}=\frac{1}{NT_{m}}.$$

## 3. Proposed Algorithm

### 3.1. Overview

Figure 3 shows the systematic process of human heartbeat and respiratory signal detection. It consists of four steps: target detection, phase extraction, signal separation and reconstruction, estimation of respiration and heartbeat rates. After performing analog-to-digital conversion on the IF signal, we first need to identify the distance range corresponding to the detected human body in the radar field of view. The distance information is obtained by range FFT, and then the range time map (RTM) is constructed according to the range distribution. After the human target is determined, the DC offset is corrected, and the arctangent demodulation phase is unwrapped by using the extended DACM algorithm. The heartbeat signal is enhanced by using the differential phase to further extract the accurate phase change information in Step 2. In Step 3, CS-OMP and RA-DWT algorithms are presented to separate and reconstruct the heartbeat and respiratory signals. Finally, we use a frequency-domain FFT algorithm and a time-domain autocorrelation algorithm to calculate the heartbeat rate and respiration rate.

### 3.2. Target Detection

Before phase extraction, we must first perform target detection to determine the location of the human being. We assume that the tested person is sitting in front of the radar and stay stably (Since the human body movement usually covers the heartbeat and breathing signals, it is difficult to detect vital sign of human body in the presence of body movements. The authors in [30] apply to solve such a problem with a Doppler radar. Although the work in [30] cannot be directly applied to the FMCW radar, it provides some research direction for the vital sign detection under body movements with FMCW radar, and we leave it for future work). These tasks will be done by the range FFT and RTM construction. A single-frame beat signal obtained after the A/D is a two-dimensional matrix, which composes of fast sampling and slow sampling. The vertical axis corresponds to the slow time sampling points constructed by the chirp frequency *N*, and the horizontal axis is the number of fast time sampling points $M_{samples}$. In order to suppress side lobe leakage, a Hamming window is added, and the range FFT vector is obtained by FFT at the fast time sampling points, so as to obtain the distance distribution from the radar field of view. In [25], the vertical axis uses a single chirp in $T_{m}$ time, and N=1. However, we experimentally found that using multiple chirps achieves a higher signal-to-noise ratio and a smaller clutter power than a single chirp. We collect data every 2 cm at the distance of 70–90 cm in front of the radar to calculate the signal peak and signal-to-noise ratio as shown in Figure 4. The results show that the clutter power in the multi-chirp mode is significantly reduced, and the signal-to-noise ratio is about 15 dB higher than that of the single-chirp mode. Therefore, we use the average range spectrum of the multi-chirp mode by calculating the mean value among columns in the two-dimensional matrix.

An RTM is constructed based on the range spectrum obtained by multi-frame coherent accumulation. The horizontal and vertical axes of the RTM represent the range and time domains, respectively, and the process is shown in Figure 5. The average range spectrum vector obtained by the range FFT is obtained by multi-frame accumulation and put into a two-dimensional matrix by rows. The horizontal axis is the distance domain obtained by FFT for fast sampling, and the vertical axis is the time domain obtained by accumulating the corresponding length of multiple frames, frame rate, $T_{m}$. The distance interval of human vital signs extraction is found on the range time matrix. This search process can be achieved by calculating the maximum average power of different distance intervals. Usually, the displacement movement brings strong reflection signals and large average power, and power changes distinguish the human chest motion displacement interval and the stationary target interval.

### 3.3. Phase Extraction

After completing the human body detection, the phase information is extracted at the distance interval where the target position is located. However, before extracting the phase information $\psi_{l}$ in a certain distance interval, it is necessary to ensure that any nonlinear distortion and interference such as false targets are removed. The DC offset of the two I/Q channels is a typical interference, which affects the accuracy of the phase extraction of vital signs. Assume that the two I/Q demodulated signals are expressed as:(11)$$B_{I}\left(t\right)=A_{I}\cos\left(2\pi f_{b}nT_{m}+\psi_{l}\right)+DC_{I},$$
(12)$$B_{Q}\left(t\right)=A_{Q}\sin\left(2\pi f_{b}nT_{m}+\psi_{l}\right)+DC_{Q},$$
(13)$$\varphi \left(t\right)=\arctan\left[\frac{B_{Q}\left(t\right)}{B_{I}\left(t\right)}\right],$$
where $A_{I}$ and $A_{Q}$ is the amplitude of the I and Q channels, respectively, $DC_{I}$ and $DC_{Q}$ are the DC offsets. There are two main reasons for the DC offset. One reason is that the external DC signal component is superimposed on the original signal. The other one comes from the local oscillator leakage and the nonlinearity of the mixer or I/Q demodulator. It is noted that the DC term contains the DC component of the original signal and the DC information of the chest displacement motion. Thus, the DC component caused by the circuit or external static objects makes the arctangent demodulation $\varphi \left(t\right)\neq \psi_{l}$. Experimental results in [31] show that the DC offset is mainly affected by circuit component defects, and the DC component of the interference needs to be corrected to obtain φ, so that $\varphi \left(t\right)=\psi_{l}$. The traditional method is to use the ADC data in the empty environment to collect the DC offset caused by the surrounding stationary targets, thermal noise, and circuits. However, this method should strictly keep the same environment during measurement and collection, which is difficult to achieve in practice. In this paper, we use a circular center dynamic DC offset tracking method. This method uses an efficient gradient descent algorithm to achieve dynamic DC offset tracking, and then performs DC offset correction [32]. Specifically, the DC offset correction is performed on the two I/Q channels, and the trigonometric function is eliminated
(14)$${\left(\frac{\left|B_{I}\left(t\right)-DC_{I}\right|}{A_{I}}\right)}^{2}+{\left(\frac{\left|B_{Q}\left(t\right)-DC_{Q}\right|}{A_{Q}}\right)}^{2}=1.$$

Considering that $A_{I}=A_{Q}=A_{R}$, we have
(15)$${\left|B_{I}\left(t\right)-DC_{I}\right|}^{2}+{\left|B_{Q}\left(t\right)-DC_{Q}\right|}^{2}={A_{R}}^{2}.$$

It can be seen that the I/Q data received will become a circle centered on the DC offset $DC_{I}$ and $DC_{Q}$. Therefore, if the center and radius are found, the DC offset of the signal can be corrected. The gradient descent algorithm is used to minimize the following optimization functions
(16)$$F\left(DC_{I},DC_{Q},A_{R}\right)=\min\sum_{k=1}^{n}\left|\left(\sqrt{{\left(B_{I}\left[k\right]-DC_{I}\right)}^{2}+{\left(B_{Q}\left[k\right]-DC_{Q}\right)}^{2}}-A_{R}\right)\right|,$$
where *k* is the number of sampling points. When the above formula achieves the minimum value, we will get the optimal result, and the DC offset and circle radius will be obtained.

It is noticed that, since the breathing and chest displacement movements is about 12 mm, which is several times of the FMCW radar wavelength (4 mm at 77 GHz). The value of the extracted phase will exceed the phase range (−π/2,π/2) if we use the arctangent demodulation technology, which will cause phase ambiguity caused by phase discontinuity and phase jump. To solve this problem, the extended DACM algorithm is used [31]. Let
(17)$$\left\{\begin{array}{c} I\left(t\right)=B_{I}\left(t\right)-DC_{I}\\ Q\left(t\right)=B_{Q}\left(t\right)-DC_{Q}, \end{array}\right.$$

The DACM algorithm turns the arctangent function into a derivative operation, then
(18)$$\frac{d}{dt}\left[\varphi (t)\right]=\frac{d}{dt}\left[\arctan\frac{Q(t)}{I(t)}\right]=\frac{I\left(t\right)Q{\left(t\right)}^{\prime }-I{\left(t\right)}^{\prime }Q\left(t\right)}{I{\left(t\right)}^{2}+Q{\left(t\right)}^{2}},$$
where $I{\left(t\right)}^{\prime }$ and $Q{\left(t\right)}^{\prime }$ are the differential forms of I(t) and Q(t), respectively. According to [31], Equation (Equation 18) is further expressed in a discrete form, and the summation is
(19)$$\varphi \left[k\right]=\sum_{k=2}^{n}\frac{I[k]\{Q[k]-Q[k-1]\}-\{I[k]-I[k-1]\}Q[k]}{I{\left[k\right]}^{2}+Q{\left[k\right]}^{2}}.$$

Although the extended DACM algorithm solves the phase ambiguity problem, the heartbeat frequency is very small and it can be easily drowned in the breathing harmonic frequencies and noise. Therefore, we perform phase difference on the phase-unwrapping signal to enhance the heartbeat signal. Differential phase is the difference between adjacent consecutive phase values, i.e., φ[k]−φ[k−1]. Such a differential phase expression enhances the heartbeat signal while suppressing the phase drift phenomenon.

### 3.4. Signal Separation and Reconstruction

The differential phase signal includes the phases of the breathing and heartbeat waveforms. After the phase difference, the heartbeat rate and the breathing rate can estimate directly, but the errors of the estimated value are large. Although the heartbeat and respiratory signals are periodic, when the harmonic frequency and noise frequency exist, the main respiratory signal is not an ideal periodic sine wave, and the heartbeat and respiratory frequency are easily drowned in the harmonic or noise. For the heartbeat frequency, the differential phase makes the heartbeat frequency more obvious; however, the harmonic peaks larger than the peak value of the heartbeat frequency are prone to occur when the breathing signal is not periodic. This will make the estimated rates of heartbeat and breathing of an intermediate frequency domain FFT method and time domain autocorrelation in Reference [22] wrong. It is worth noting that such problems cannot be avoided when estimating the heartbeat rate and respiration rate [25]. Therefore, we provide two methods for the separation and reconstruction of heartbeat and respiratory signals.

#### 3.4.1. CS-OMP Algorithm

Since heartbeat frequency is 0.8–2 Hz, and the breathing frequency is 0.1–0.5 Hz, the heartbeat and breathing signals can be separated using bandpass filters. In this paper, we first design two bandpass filters (serially-cascaded Bi-Quad IIR filter) to separate the heartbeat and respiratory signals according to these two frequency bands. The sampling frequency is 20 Hz, and the pass-band gain and stop-band gain are, respectively, 1 dB and 40 dB. The differential phase signal is passed through the two designed band-pass filters, and the heartbeat and breathing signals are separated as shown in Figure 6. As shown in the spectrum in Figure 6b,d, there are always harmonic and noise frequencies in the spectrums of the separated two bands that affect the estimation of heartbeat rate and respiratory rate. Therefore, we propose a CS-OMP algorithm to reconstruct the heartbeat and breathing signals, and suppress harmonic interference and noise.

According to the compressed sensing theory, if the signal is sparse in a transformation domain, an observation matrix unrelated to the transformation basis can be used to project the transformed high-dimensional signal into a low-dimensional space. Therefore, by collecting a small number of signal projection values, the accurate or approximate reconstruction of the signal can be achieved [33]. Since the sparsity of heartbeat and breathing signals has been proven in [34], the compression and reconstruction of the heartbeat and breathing signals in this paper can be implemented by the compressed sensing theory. Then, the sparsity of the heartbeat or breathing signals is given by [35]
(20)x=Γ(α+w),
where $\Gamma =\{\Gamma_{1},\Gamma_{2},\Gamma_{3},\cdots ,\Gamma_{P}\}$ is the orthogonal transform basis in the frequency domain, α is the weight coefficient of P×1, and *w* is the noise. If the signal *x* has only *K* (K<<P) non-zero coefficients on a transformation basis Γ, then, Γ is called the sparse basis of *x*, and *K* is the number of sparse signals. The smaller the *K*, the greater the signal sparseness. The original signal is projected onto the measurement matrix $\Phi =[\phi_{1},\phi_{2},\phi_{3},\cdots ,\phi_{P}]$ of M×P by retaining the *K* important characteristic components in the non-adaptive linear projection *y*. Then, the non-adaptive linear projection value of the signal *x* is
(21)$$y=\Phi x=\Phi \Gamma \left(\alpha +w\right)=A_{CS}\alpha +Z,$$
where $A_{CS}=\Phi \Gamma$ is the perception matrix, Z=ΦΓw is the projection value of noise. Especially when there is no noise, $y=A_{CS}\alpha$. When the measurement matrix M<<P, reconstructing *x* from *y* is an ill-posed problem. However, since the signal *x* is sparse, $L_{1}$-norm can be applied to transform the problem as a convex optimization problem [33]. Therefore, the ill-posed problem is transformed into the following optimization one
(22)$$\arg\min{∥\alpha ∥}_{1}\;s.t.{∥A_{cs}\alpha -y∥}_{2}\leq \varepsilon ,$$
where ${∥\alpha ∥}_{1}$ is the $L_{1}$-norm of α , and ε is the noise boundary in the data. By solving Equation (Equation 22), we can obtain the sparse coefficient vector $\hat{\alpha }$, and then the reconstructed signal is $\hat{x}=\Gamma \hat{\alpha }$, where $\hat{\alpha }$ is the optimal solution. We use the OMP algorithm [36] to reconstruct signal *x* from $\hat{x}=\Gamma \hat{\alpha }$. To obtain a noiseless heartbeat or breathing signal (K=1 in this paper), important feature components in *y* are retained and reconstructed using OMP. It is noted that reconstruction failure occurs during the reconstruction process due to the fact that the reconstructed signal may not be noiseless. Therefore, in order to ensure that the reconstructed signal is the heartbeat or breathing signal after denoising, a restriction condition is set. The reconstructed signal is output only when the peak value of the reconstructed signal spectrum equals the peak value of the original one. Then, the CS-OMP algorithm can be described as Algorithm 1.
**Algorithm 1.** CS-OMP algorithm for breathing signal.**Input:** perception matrix $A_{CS}=\Phi \Gamma$, Observation vector *y*, Signal spectrum peak frequency $f_{max}$, and the noise boundary ε.**Output:** 1-sparse reconstructed signal for *x*, $\hat{x}$.1:Initialization residual $r_{0}=y$, K=1, index matrix $\Lambda_{0}=\Phi_{0}=\emptyset$.2:Find the index $\lambda_{K}$ corresponding to the residual $r_{K}$, and the maximum value of the column inner product of the measurement matrix such that $\lambda_{K}=\underset{j=1,\cdots ,N}{\arg\max}\left|\langle r_{K-1},\phi_{j}\rangle \right|$;3:Update index matrix $\Lambda_{K}=\left[\Lambda_{K-1},\lambda_{K}\right]$ and $\Phi_{K}=\left[\Phi_{K-1},\phi_{\lambda_{K-1}}\right]$;4:Calculate the least squares solution, ${\hat{\alpha }}_{K}=\underset{\alpha_{K}}{\arg\min}{∥y-A_{CS_{K}}\alpha ∥}_{2}$;5:Update $r_{K}=y-A_{CS}{\hat{\alpha }}_{K}$;6:If ${∥r_{K}∥}_{2}>\varepsilon$, return to Step 1. Otherwise, reconstruct the signal *x* by using $\hat{x}=\Gamma {\hat{\alpha }}_{K}$;7:If the reconstructed signal spectrum is $f=f_{max}$, stop, output $\hat{x}$. Otherwise, return to Step 2.

The proposed CS-OMP algorithm will get better results for the respiration signal reconstruction, and the accuracy of its corresponding respiration rate will be improved. This is due to the fact that the respiratory frequency always has the maximum peak in its spectrum. For the heartbeat spectrum, its maximum peak may not necessarily correspond to the heartbeat frequency. As mentioned before, breathing harmonics, noise, and body movements may have larger peaks than the real heartbeats. Fortunately, the heartbeat frequency is always the maximum or sub-peak of its spectrum and the heartbeat rate estimation method will be given later.

#### 3.4.2. RA-DWT Algorithm

Heartbeat and respiration signals belong to non-stationary signals, which are very important for the frequency domain characteristics of non-stationary signals at any time. The wavelet transform has multi-resolution characteristics, and the selection of appropriate scale factor and telescopic coefficient can be used to represent the local characteristics of signals in time domain and frequency domain by the mother wavelet. The discrete wavelet transform is used to perform wavelet decomposition on the differential signal. The discrete wavelet transform is described as [37]
(23)$$y_{l}\left[n\right]=\sum_{k=-\infty }^{\infty }\varphi [k]\cdot g[2n-k],$$
(24)$$y_{h}\left[n\right]=\sum_{k=-\infty }^{\infty }\varphi [k]\cdot h[2n-k],$$
where φ[k] is the discrete differential phase signal, *k* is the number of sampling points, and *n* is the number of sampling data. g[2n−k] and h[2n−k] are respectively the low-pass and high-pass filter decomposition coefficients. The $y_{l}\left[n\right]$ and $y_{h}\left[n\right]$ are the approximate and detail coefficients after decomposition, respectively. After the finite order decomposition of φ[k], it can be reconstructed by DWT decomposition
(25)$$S\left[k\right]=\sum_{i=a}^{b}y_{l}\left[n\right]\cdot G\left[2n-k\right]+\sum_{m=a}^{b}y_{h}\left[n\right]H\left[2n-k\right],$$
where *a* is the scale factor, *b* is the displacement factor, G[2n−k] and H[2n−k] are low-pass and high-pass reconstruction coefficients for scaling and wavelet functions, respectively [38]. The approximate and detailed coefficients after each order of decomposition can reconstruct the corresponding two sets of the signals, and obtain the desired frequency interval signal according to the breathing and heartbeat frequency ranges
(26)$$S_{BR}\left[k\right]=\sum_{k=-\infty }^{\infty }S[k]\cdot g\left[2n-k\right],$$
(27)$$S_{HR}\left[k\right]=\sum_{k=-\infty }^{\infty }S[k]\cdot h\left[2n-k\right].$$

In this paper, “db5” wavelet is selected to perform wavelet decomposition on the input discrete differential signal φ[k] according to the two expected frequency intervals of heartbeat and respiration signals. After obtaining the heartbeat and breathing signals in the desired frequency range, the signals are quantified by a threshold value. The rigrsure adaptive soft threshold Stein’s unbiased risk estimation rule is selected for noise reduction [39], and then the heartbeat and breathing signals are reconstructed. After the FFT of the noise-reconstructed signal is obtained, the peak value of the spectrum can be used as the estimation of the respiration or heartbeat frequency. However, there may be multiple decomposed and reconstructed signals falling in the desired frequency range. Therefore, a confidence metric, *W*, is defined for each reconstructed signal, representing the ratio of the signal power at the maximum peak (and some surrounding frequencies) to the remaining frequency intervals in the spectrum:(28)$$W=\frac{\int_{f_{\max}-\hbar /2}^{f_{\max}+\hbar /2}\left|P(f)\right|df}{\int_{0}^{\infty }\left|P(f)\right|df-\int_{f_{\max}-\hbar /2}^{f_{\max}+\hbar /2}\left|P(f)\right|df},$$
where $\left|P(f)\right|$ is the signal power at frequency *f*, and $\hbar =F_{slow}/M_{samples}$ is the frequency resolution, calculated as the ratio of the slow time sampling rate $F_{s\_slow}$ to the number of FFT sampling points. The greater the confidence measure *W* is, the more accurate the peak frequency of the corresponding reconstructed signal is as a heartbeat or breathing frequency estimate. It is noted that, when the confidence level is less than a certain threshold value, it indicates that the peak frequency of the corresponding spectrum is unreliable.

### 3.5. Estimation of Respiration and Heartbeat Rates

The respiration and heartbeat rates can be calculated using the FFT method after the CS-OMP and RA-DWT reconstruction. However, as mentioned before, the method of heartbeat rate estimation after reconstruction using the CS-OMP algorithm needs to be redefined. We find out all the peaks in the spectrum of reconstructed signal and keep them in 0.8–2 Hz, and remove the respiratory harmonic peak. Then, the frequency corresponding to the peak of the reconstructed signal is counted and the frequency weight coefficient is defined as $\xi_{i}=m/q$, where *m* is the number of occurrences of the corresponding frequency of reconstruction, and *q* is the number of reconstructions. Then, the heartbeat rate is
(29)$${\bar{f}}_{h}=\sum_{i=1}^{N}f_{i}\xi_{i}.$$

For comparison, both the CS-OMP and RA-DWT reconstruction can be combined with the autocorrelation method to estimate the heartbeat and respiration rates. For breathing and heartbeat with a limited period signal, the corresponding autocorrelation function will gradually become zero, and a peak corresponding to the multiple of the basic period will appear in the time:(30)$$R_{xx}\left(m\right)=\rho \sum_{k=-\infty }^{k=\infty }x\left(k+m\right)x^{*}\left(k\right),$$
where ρ is the normalization factor, *m* is the lag time, and ${}^{*}$ is the conjugate multiplication. The autocorrelation functions of the separated and reconstructed heartbeat and respiration signals are calculated. Then, the corresponding time of the autocorrelation peak in the heartbeat and breathing time interval is estimated, and the inverse number is the corresponding breathing and heartbeat frequency.

The calculation of real-time heartbeat and breathing rate is given by
(31)$$\gamma =\frac{t_{convert\_hz\_bpm}\cdot R_{numPeaks}\cdot F_{s}}{L_{circularBufferSize}},$$
where $t_{convert\_hz\_bpm}$ is the measurement duration, $L_{\mathrm{circularBufferSize}}$ is the radar buffer size, $R_{numPeaks}$ is the number of signal crest when the data length of the buffer area is $L_{\mathrm{circularBufferSize}}$. Specifically, $t_{convert\_hz\_bpm}=60$ s, the heartbeat and respiration rates are calculated as the number of heartbeats and breaths per minute.

## 4. Experimental Results and Discussion

### 4.1. Experimental Parameters

This paper uses Texas Instruments (TI)’ millimeter-wave radar AWR1642 (State of Texas, US), which is an integrated single-chip FMCW radar sensor operating in the 77 to 81 GHz frequency band [40]. The experimental application scenario is as shown in Figure 7a. The person to be tested sits in front of the radar and remains stationary, wearing an airflow sensor and a smart bracelet Mi 3. The airflow sensor is used to get the breathing rate and the smart bracelet Mi 3 is adopted to capture the heartbeat rate. It is noted that, although the person sits in front of the radar and remains stable, there still exists noise and micro-motion interference from the human body itself and the nearby environment. AWR1642 integrates TI’s high-performance C674x DSP subsystem for radar signal processing, which can be programmed on the radar board. However, to observe the experimental results and data analysis, we hope to obtain the original data for algorithm verification. Therefore, human heartbeat and respiratory data are collected by DCA1000 [41] combined with AWR1642 in streaming data mode.

AWR1642 adopts two transmitting and four receiving one-dimensional linear array antenna layout, and uses time division multiplexing (TDM) mode to alternately transmit antenna TX1 and TX2 frequency modulation pulses to obtain eight virtual antenna arrays, as shown in Figure 7b. In our experiment, a single TX/RX antenna pair can implement single-person heartbeat and respiration detection. However, in order to get a longer detection distance, we increase the signals gain by using two TXs. We alternately send a single FM pulse with duration time, $T_{c}=50\;\mu$s, and the pulse interval idle time is 7μs, and the sawtooth frequency modulation slope is S=70 MHz/μs, as a result, the frame rate is $T_{m}=50$ ms. Considering that the observation time contains at least two cycles of breathing or heartbeat signals, to better analyze the heartbeat rate and breathing rate per minute, we set the observation time T=50 s. Thus, the frame length $C(C=T/T_{m})$ is 1000. The specific frequency chirp form is shown in Figure 7, and the specific radar parameter settings are listed in Table 1.

Combining Equations (Equation 7) and (Equation 9) with (Equation 10), the actual distance measurement accuracy and frequency vibration measurement range of the AWR1642 radar are calculated in a complex 1× mode, as shown in Table 2. The breathing frequency is in the range of 0.1–0.5 Hz, and the heartbeat frequency is in the range of 0.8–2 Hz. The measurement accuracy range in Table 2 shows that the radar parameter settings are sufficient to achieve non-contact human heartbeat and breathing detection.

Since we use the DCA1000 to capture the raw radar data, the data will become a non-interleaved data format [41] after ADC. The ADC data need de-ping-pong to build a complex signal, and then serial-parallel conversion is performed to correspond the TX/RX antenna pair.

### 4.2. Experimental Results

#### 4.2.1. Target Detection

The RTM image constructed according to the flow of Figure 5 is shown in Figure 8a. Figure 8b shows the single frame (single chirp) range information, and there are reflections from different targets in the range of 0.6–0.9 m. During the experiment, the human body is always sitting in front of the radar (see Figure 7a), which means that the target body moves up and down only because of heartbeat and breathing movement. As mentioned before, the moving target always brings stronger signal power, which is reflected in the brighter part of the image. By comparing the average power of unit range interval, the maximum power interval caused by heartbeat and respiration is selected, which is shown as 0.7–0.84 m in Figure 8a. The vibration FFT can be used to verify the selected distance interval, and the second FFT is performed on the phase-unwrapping row to obtain the vibration FFT, as shown in Figure 8b. The range corresponding to the vertical axis is about 21 times/min, and this is caused by the predominant breathing vibration, which falls in the 0.1–0.5 Hz (6–30 beats/min) interval, and it is proved that the distance interval we choose is correct. Moreover, we observe that 0.02–0.1 m in Figure 8a has a target, which is caused by the spectral leakage because of the antenna coupling phenomenon. It can be removed after DC correction, which is reflected in the vibration FFT of Figure 8c. Except for the body motion vibration caused by heartbeat and breathing, the vibration frequency no longer exists in the other sections.

#### 4.2.2. Phase Extraction of Heartbeat and Breathing

The result after DC offset correction can also be seen in the time domain and the corresponding frequency domain. The real part of the complex data of the collected radar is taken as the horizontal axis, and the imaginary part is taken as the vertical axis. The circular center tracking algorithm is applied for DC offset correction, and the results are shown in Figure 9a. The green points are the raw data, and the blue ones are the data after DC offset correction. It can be clearly seen that the center and the blue points of the green circle (shifted origin and before DC cor. in Figure 9a are moved to the blue circle with center Origin. Figure 9b shows the comparison before and after the time-domain waveform correction. It can be seen from Figure 9c that the DC offset dynamic tracking of the circle center can effectively suppress the DC offset caused by non-thoracic displacement motion.

Figure 10a shows the phase after arctangent demodulation, and phase jumps and discontinuities appear in the figure. The extended DACM algorithm is used to obtain Figure 10b. After phase unwrapping, the time domain waveform is closer to a sine wave. It is learnt from Figure 10b that the heartbeat frequency becomes weak and it is easily submerged in the fundamental harmonic and noise. It can be seen that, in the spectrum of Figure 10c after the phase difference, the heartbeat frequency becomes more obvious. It is noted that the large waveform in Figure 10b is caused by the respiratory vibration, whereas the vibration caused by the chest displacement movement is always superimposed on the top of the waveform. In the frequency spectrum, there are tiny frequencies close to 0, and they are caused by small tremors such as muscles or bones.

#### 4.2.3. Separation and Reconstruction of Heartbeat and Breathing Signals

The CS-OMP algorithm is proposed to construct the phase signal in Figure 11. The red line is the original heartbeat and breathing signal after bandpass filtering, and the blue one is the reconstructed signal. It is obviously learnt from the right two figures of Figure 11 that the reconstructed signal is an ideal periodic sinusoidal signal composed of a single frequency, and there is no other interference for the frequency estimation of the heartbeat and breathing.

The heartbeat and respiration signals in the differential phase signal can be obtained from coarse to fine according to the confidence formula. The peak frequency and confidence value of the reconstructed signal using “db5” under different order decompositions are given in Table 3. It can be learnt that, at the 4th order, the peak frequency of the reconstructed heartbeat signal is 1.46 Hz, and the corresponding confidence value is 0.4155. This confidence value is the largest in the frequency range of 0.8–2 Hz; as a result, 1.46 Hz will be used as the estimated heartbeat frequency. Similarly, 0.34 Hz at the 6th order decomposition is the breathing rate frequency. Then, we can get the heartbeat signal after the 4th order wavelet decomposition, and the breathing signal is obtained after 6th wavelet decomposition. Therefore, to show the multiple decomposition process of the phase differential signal, we give the reconstructed breathing and heartbeat signals in Figure 12. In Figure 12, the blue line is the reconstructed breathing signal with approximate coefficients, and the green one is the reconstructed heartbeat signal with detailed coefficients. Ideally, the heartbeat and breathing signals are ideal periodical, and the reconstructed breathing and heartbeat signals after 6th and 4th order wavelet decomposition are not strictly periodical because of the noise and interference. To preserve the ideal periodicity of the heartbeat and breathing signals and the correctness of the reconstructed signals, the adaptive soft threshold with rigrsure is further used for noise reduction. The red lines in Figure 12 show the heartbeat and breathing signals of the noise reduction results. It can be seen that the denoised results are much better than the original ones.

Figure 13 shows the frequency spectrum of the breathing (in Figure 13a) and heartbeat signals (in Figure 13b) reconstructed after the 6th and 4th order wavelet decomposition and noise reduction. The blue line represents the spectrum of the heartbeat and breathing signals after bandpass filtering. It can be seen that the interference frequency is significantly reduced after wavelet decomposition and noise reduction.

In order to observe the respiration waveform and heartbeat waveform after reconstruction, the observation time window is set to 50 s. In addition to the frequency domain estimation method, the autocorrelation function is used to estimate the respiratory and heartbeat rates in the time domain. The experimental results are shown in Figure 14.

### 4.3. Performance Comparison

To further verify the accuracy and effectiveness of the proposed algorithms, we compare our work with the filtering method that proposed in [25]. The filtering method uses two bandpass filters to separate the breathing and heartbeat signals, and find the largest spectral values in the two bands as the corresponding estimated breathing and heartbeat rate. In this paper, the radar is used to collect data for five minutes on five adult males and five adult females, respectively. For the collected offline data, the data length of one minute is taken for respiratory rate and heartbeat rate estimation. The respiratory rate and heartbeat rate estimation in frequency domain FFT and time domain autocorrelation are used to obtain Table 4 and Table 5. In the tables, the mean standard deviation (MSD) and the Pearson correlation coefficient (PCC) are used for performance comparison. The MSD is defined as:(32)$$\sigma_{e}=\frac{\sqrt{\frac{1}{n}\sum_{i=1}^{n}{\left(e_{i}-\bar{e}\right)}^{2}}}{\bar{d}},$$
where $e_{i}$ is the difference between the corresponding reference heartbeat rate (or respiration rate) and the estimated value; $\bar{e}$ is the average difference; and $\bar{d}$ is the data average rate. Reference heartbeat rate and respiratory rate are provided by contact sensing device. PCC is defined as [25]
(33)$$PCC=\frac{\sum_{i=1}^{N}\left(d_{ri}-{\bar{d}}_{ri}\right)\left(d_{si}-{\bar{d}}_{si}\right)}{\sqrt{\sum_{i=1}^{N}\left(d_{ri}-{\bar{d}}_{ri}\right)}\sum_{i=1}^{N}\left(d_{si}-{{\bar{d}}^{2}}_{si}\right)},$$
where $d_{ri}$ and $d_{si}$ are the *i*-th estimated rate (respiratory or heartbeat rate) and contact sensor reference rata, respectively. The value of PCC ranges from [0–1], where 0 indicates that the two are not related, and 1 indicates that the two are completely linearly related. In Table 4 and Table 5, when the band-pass filtering method is used in conjunction with the frequency FFT and time-domain autocorrelation methods, the MSD is higher and the PCC is reduced due to the passband noise and harmonic interference. The use of CS-OMP and RA-DWT method while ensuring lower MSD, PCC has also been correlated higher guaranteed. It is observed that the MSD and PCC of the two methods of CS-OMP are always the same because CS-OMP is reconstructed as a single frequency periodic sine wave, and both its frequency domain and time domain autocorrelation are corresponding to a single frequency or period. We also learn from Table 4 and Table 5 that the accuracy of heartbeat rate is improved by about 7%, and the accuracy of breathing rate is improved by about 4%.

Figure 15a,b respectively show the real-time heartbeat rate and breathing rate obtained by the RA-DWT method using the contact sensing devices (smart bracelet Mi 3 and Airflow sensor) and AWR1642 radar sensor of male 5 within 10 min.

### 4.4. Discussion

After applying the circular center dynamic tracking algorithm to correct the DC offset, the extended DACM allows us to obtain effective heartbeat and breathing-related phase information. Two methods, CS-OMP and RA-DWT, are proposed to process the extracted differential phase signal. The heartbeat and breathing waveform reconstructed by the CS-OMP algorithm shows an ideal periodic sine wave, and its corresponding autocorrelation function is also a periodic sine signal with no other interference frequencies. The heartbeat and breathing signals reconstructed by RA-DWT reduce the irregular waveform phenomenon, and effectively reduce the noise and harmonic interference while retaining the time-frequency characteristics of the original signal. Compared with the general band-pass filtering method, the accuracies of heartbeat rate and respiration rate are improved.

The heartbeat signal reconstructed by CS-OMP may be noise that is close to the true heartbeat frequency, and there is still a certain estimation bias in using the statistical value of the weighting coefficient to approximate the true value. It is worth noting that the RA-DWT in Table 4 and Table 5 has a higher PCC and a lower MSD than the CS-OMP method, but this does not mean that the latter is worse than the former. In Figure 15a,b, the PCC for the real-time heartbeat rate and the real-time respiration rate are compared with the results in Table 4 and Table 5 of the offline collected data processing, and the RA-DWT method is reduced instead. Therefore, for offline data processing results, the PCC obtained by RA-DWT is a little higher than CS-OMP, but, for real-time processing, due to hardware equipment and other influencing factors, the accuracy of RA-DWT detection may be reduced. In our experiments, the effect of the two methods is actually equivalent, and more experiments are needed to verify which one is better. However, compared with the traditional method of band-pass filtering in [25], the MSD of CS-OMP and RA-DWT are always lower, and both heartbeat rate and breathing rate of the PCC value can be guaranteed at about 93%.

## 5. Conclusions

In this paper, a 77 GHz FMCW radar was applied to obtain the breathing and heartbeat signals via extracting the radar IF phase information. This paper systematically introduced the radar data signal processing flow and parameter configuration. Two methods for separating and reconstructing the heartbeat and breathing signals were proposed, and the results were compared with those of the contact sensor devices (smart bracelets and airflow sensors). Experimental results showed that both the proposed methods effectively suppressed the influence of noise and harmonic interference. The detection accuracies of both heartbeat rate and respiration rate were higher than 93%, which proved the feasibility and effectiveness of FMCW radar in remote vital signs monitoring. The methods proposed in this paper achieve high accuracy in static human life detection, and a more practical scenario with human body movement is interesting, and we leave it for future work.

## Figures and Tables

**Figure 1 sensors-20-02999-f001:**
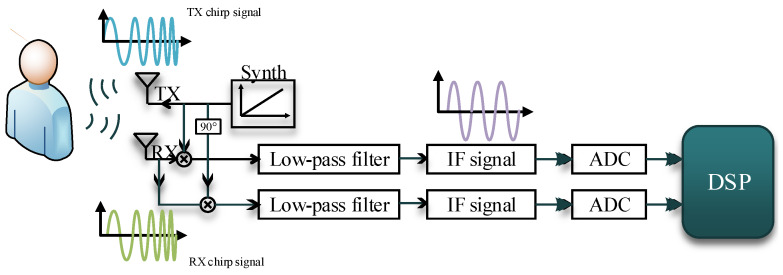
FMCW radar system block diagram.

**Figure 2 sensors-20-02999-f002:**
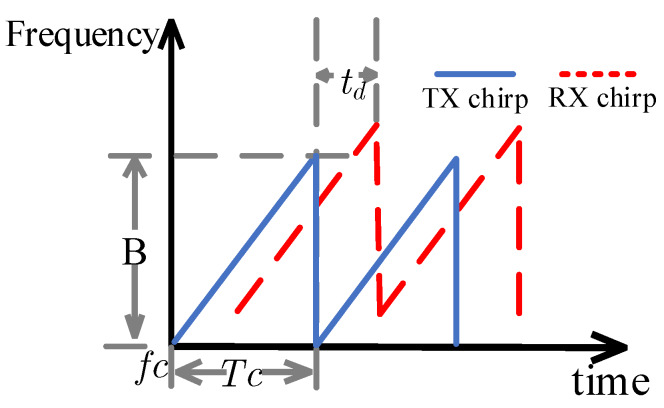
Time dependence of transmitted and received signals (frequency as a function of time).

**Figure 3 sensors-20-02999-f003:**
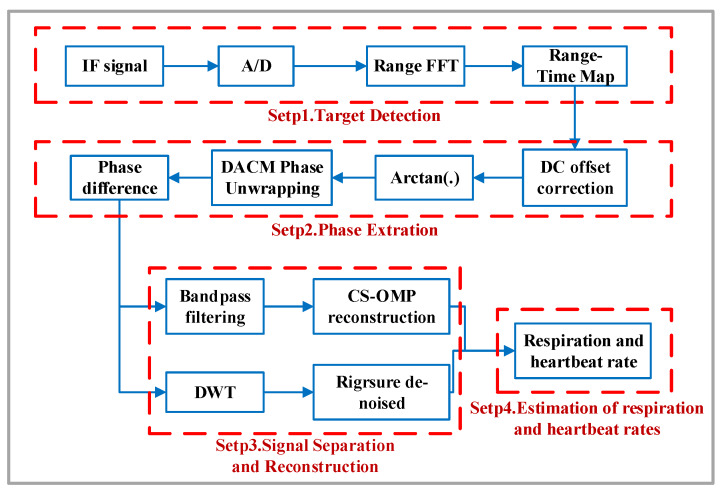
Human heartbeat and breathing signal detection flowchart.

**Figure 4 sensors-20-02999-f004:**
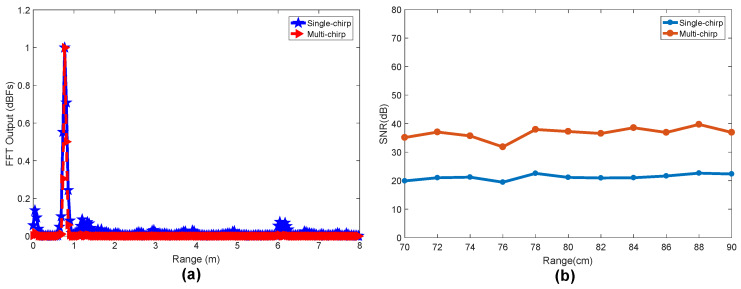
Single chirp and multiple chirps comparison: (**a**) range spectrum; and (**b**) signal-to-noise ratio.

**Figure 5 sensors-20-02999-f005:**
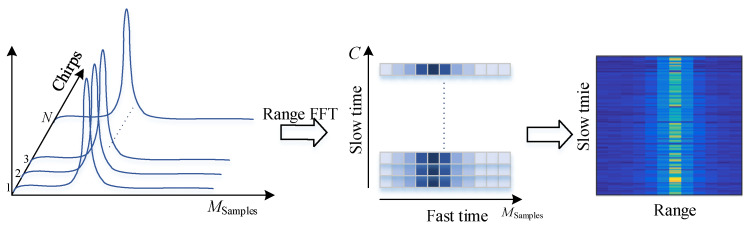
RTM construction.

**Figure 6 sensors-20-02999-f006:**
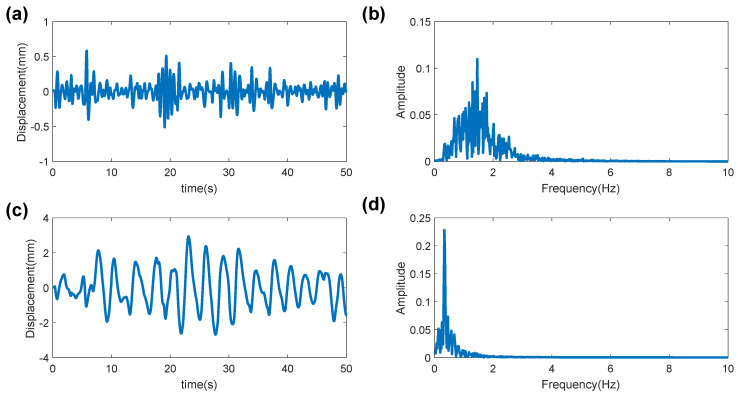
Waveform and spectrum of separated signal: (**a**) heartbeat waveform; (**b**) heartbeat spectrum; (**c**) respiratory waveform; and (**d**) respiratory spectrum.

**Figure 7 sensors-20-02999-f007:**
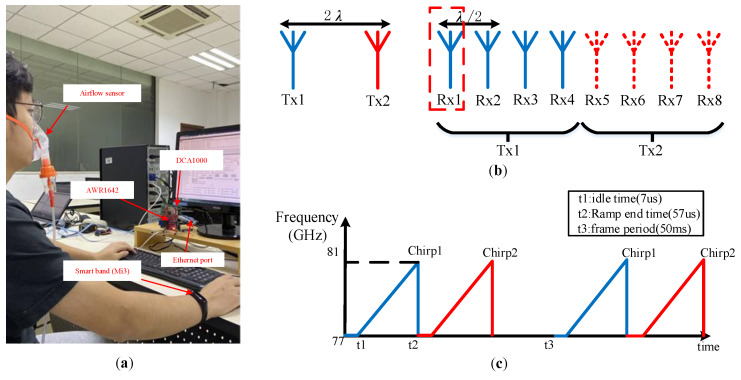
Experimental platform: (**a**) experimental scene; (**b**) radar antenna layout; and (**c**) radar chirp parameters setting.

**Figure 8 sensors-20-02999-f008:**
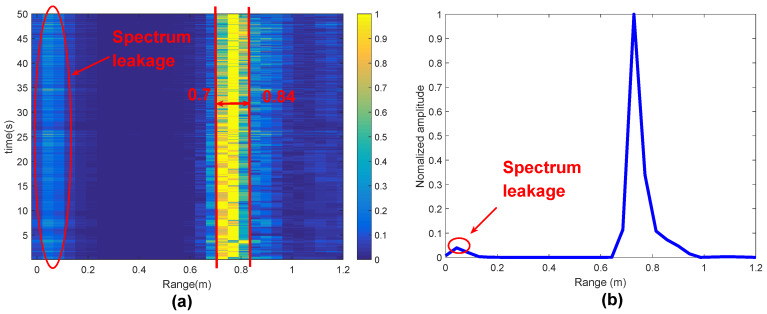
Target Detection: (**a**) RTM; (**b**) range FFT; and (**c**) vibration-range.

**Figure 9 sensors-20-02999-f009:**
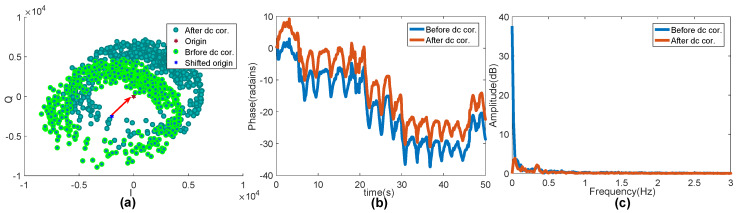
DC offset correction: (**a**) center-dynamic DC offset tracking correction; (**b**) time-domain; and (**c**) frequency domain.

**Figure 10 sensors-20-02999-f010:**
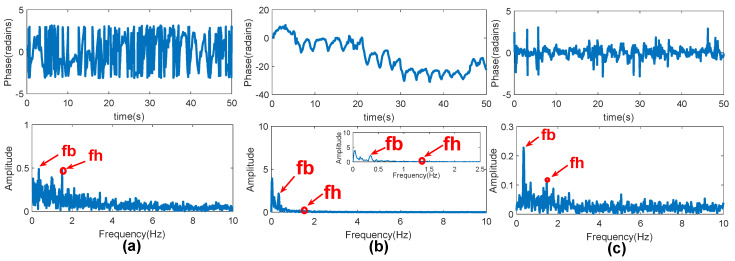
Phase extraction: (**a**) phase waveform and its corresponding spectrum after arctangent demodulation; (**b**) phase waveform and its spectrum after phase unwrapping; and (**c**) phase waveform after phase difference and its spectrum.

**Figure 11 sensors-20-02999-f011:**
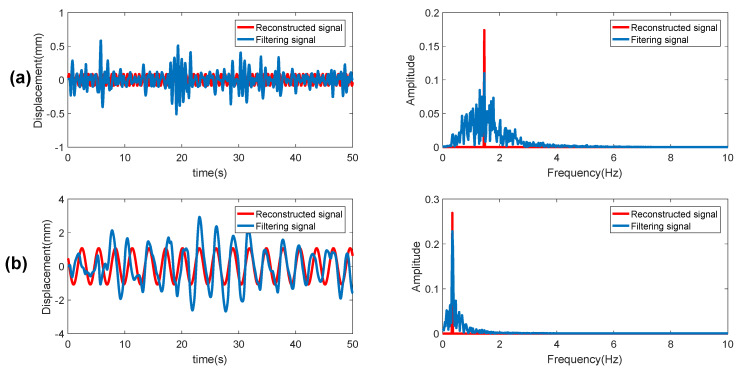
CS-OMP reconstructed signal: (**a**) heartbeat signal and its spectrum; (**b**) respiration signal and its spectrum.

**Figure 12 sensors-20-02999-f012:**
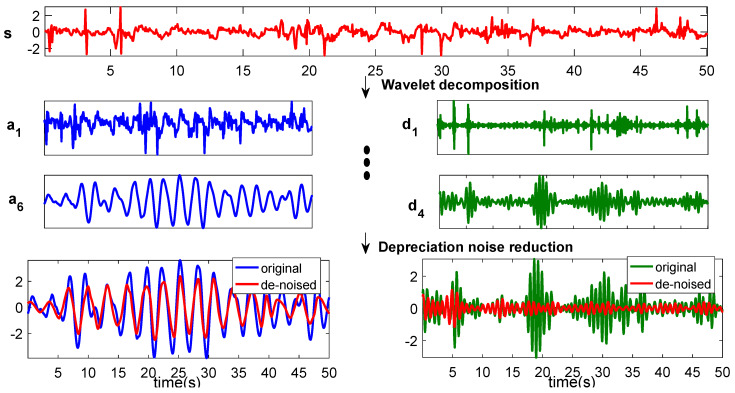
Wavelet decomposition and noise reduction reconstruction results.

**Figure 13 sensors-20-02999-f013:**
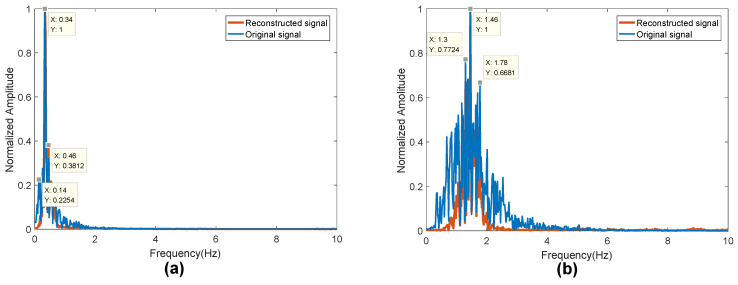
Wavelet decomposition spectrum: (**a**) respiratory signal; and (**b**) heartbeat signal.

**Figure 14 sensors-20-02999-f014:**
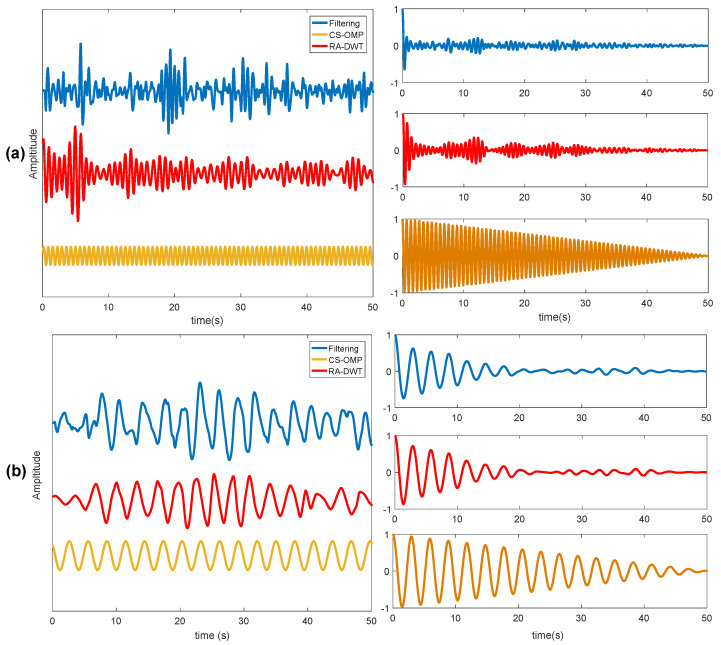
Separate reconstructed waveform: (**a**) heartbeat waveform and its autocorrelation; and (**b**) breathing waveform and its autocorrelation.

**Figure 15 sensors-20-02999-f015:**
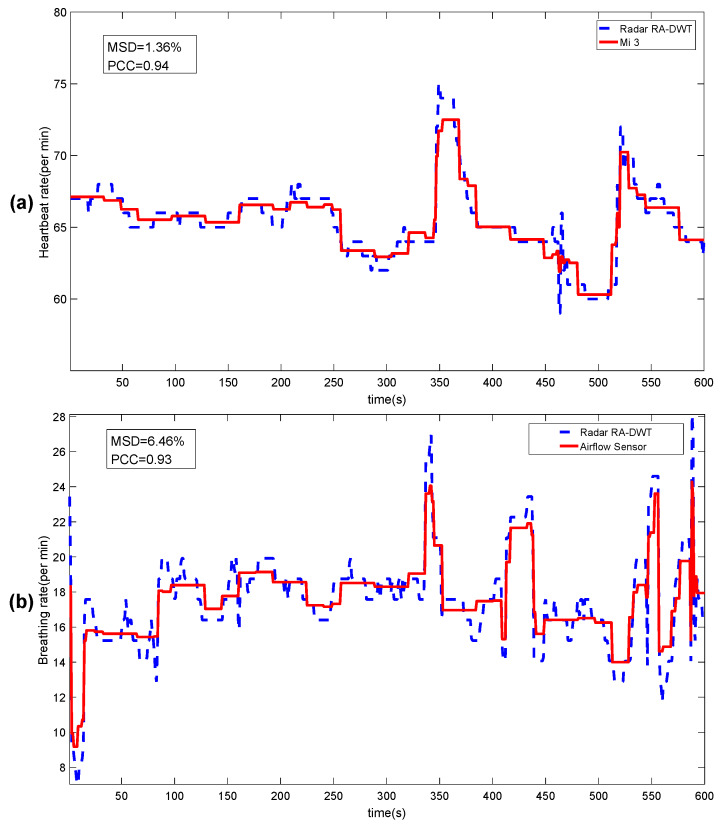
Real-time comparison: (**a**) heartbeat rate; and (**b**) breathing rate.

**Table 1 sensors-20-02999-t001:** Radar parameter setting.

Parameter	$T_{c}$	$T_{m}$	*S*	$F_{s\_\mathrm{adc}}$	$F_{s\_\mathrm{slow}}$	$M_{\mathrm{samples}}$	C
Value	50μs	50 ms	70 MHz/μs	4 Msps	20 Hz	200	1000

**Table 2 sensors-20-02999-t002:** Distance and vibration frequency detection range.

Parameter	Range (m)	Vibration Frequency (Hz)
Max	6.7	10
Min	0.0335	0.020

**Table 3 sensors-20-02999-t003:** Confidence value of the reconstructed signal under different order decompositions.

Decomposition Level	$f_{n\_\max}$	$f_{d\_\max}$	W1	W2
2	0.46	-	0.1217	-
3	0.46	1.78	0.2092	0.0614
4	0.46	1.46	0.4032	0.4155
5	0.28	1.3	0.3915	0.3868
6	0.14	0.34	0.3817	0.9229
7	-	0.14	-	0.5384

**Table 4 sensors-20-02999-t004:** Heartbeat rate comparison between different methods.

Subjects	Mi 3 (Reference Heartbeat Rate)	AWR1642 Radar Sensor					
		Filtering [25]		CS-OMP		RA-DWT	
		FFT	Auto- Correlation	FFT	Auto- Correlation	FFT	Auto- Correlation
Male 1	88	92	85	92	92	87	85
Male 2	77	85	75	73	73	79	77
Male 3	74	80	79	74	74	75	74
Male 4	60	75	65	58	58	60	58
Male 5	67	80	77	65	65	65	62
Female 1	81	97	83	83	83	82	80
Female 2	71	82	76	78	78	75	73
Female 3	85	93	88	87	87	89	82
Female 4	73	86	82	72	72	70	70
Female 5	82	95	90	80	80	85	83
MSD (%)	-	4.63	5.43	4.38	4.38	3.16	2.92
PCC (%)	-	88	86	95	95	97	97

**Table 5 sensors-20-02999-t005:** Breathing rate comparison between different methods.

Subjects	Airflow Sensor (Reference Breathing Rate)	AWR1642 Radar Sensor					
		Filtering [25]		CS-OMP		RA-DWT	
		FFT	Auto- Correlation	FFT	Auto- Correlation	FFT	Auto- Correlation
Male 1	17	21	18	17	17	17	17
Male 2	19	24	21	20	20	19	19
Male 3	23	26	25	22	22	25	23
Male 4	16	19	17	15	15	17	15
Male 5	22	21	19	20	20	22	21
Female 1	23	28	25	25	25	23	22
Female 2	18	19	16	18	18	19	17
Female 3	28	32	28	28	28	28	25
Female 4	24	28	27	26	26	25	25
Female 5	25	27	23	23	23	25	24
MSD (%)	-	7.70	9.43	6.78	6.78	3.21	5.09
PCC (%)	-	90	88	94	94	98	96

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
