# Peer review of "Remote Monitoring of Human Vital Signs Based on 77-GHz mm-Wave FMCW Radar"

_sensors, 2020, doi:10.3390/s20102999_

Round 1

Reviewer 1 Report

This is an interesting topic and the results seem to be effective. Although each step uses rather standard techniques, the combination of them to resolve an important problem is plausible. It is recommended to publication after a minor revision by addressing the following comments.

1. In the abstract, "higher than the existed radar-based method by more than by  13\%", the authors should provide the references for the number.

2. The use of FFT to obtain human vital signs using UWB radar was considered before and should be included in the references, e.g.,

    Li et al. , Through-wall detection of human being's movement by UWB radar, IEEE Geoscience and Remote Sensing Letters, Vol.9 (2012) Iss. 6 1079-1083.    

3. Page 1, Line 2, delete "without wearing or contacting any equipments"

4. Page 1, Line 4, change "a focus" to "attention"

5. Page 4, Line 131, can the author give an estimation of the speed of human chest motions due to heartbeat and breathing?

6. Do the height, weight, or other elements affect the performance of the proposed methods?

Author Response

Dear Reviewer,

Thanks for your comments. We have responsed your comments point by point, and the response letter has been attached. Thanks for your time and considerations.

Best

The authors

Reviewer 2 Report

The paper proposed new algorithms for detection of heartbeat and respiration rate from FMCW radar. The use of FMCW radar for vital sign monitoring has been already widely investigated and the contributions of the current work appears to be on the algorithm side. However, in addition to the novelty, there are some major concerns regarding the current work.

1- The experimental setup is not clear with no enough description of how the measurements were performed. Did the subjects performed any movements? the main problem using radars is when the subjects move, have the authors tried exploring the accuracy of their method when movements happen? The usage of the proposed algorithm in such cases only can prove its usability and accuracy.

2- Given the subjects were stationary, the average heartbeat and respiration rates are very close among the subjects. The proposed method should be tested in difference scenarios where the heart rate and respiration rate have been elevated. 

3- Why are the original and denoised signals are so similar in Figure 12?

Author Response

(The authors gave the same response as above.)

Reviewer 3 Report

Figures 2 and 8 should be revised for better clarity.

In line 130, mention is made that the human chest motion does not exceed 103 m/s. This number needs to be confirmed as well as some discussion added as to its significance to the paper and procedures presented.

In line 33, mention is made of several possible uses of non-contact radar sensors that use FMCW.  Missing are lower frequency Oceanographic HF Radars used to measure ocean current and direction for various surface tracking uses including naval search and rescue operations.  Many publications are available which cover this topic.

Author Response

Dear Reviewer,

We have responsed your comments point by point, and the response letter have been attached. Thanks for your time and considerations.

Best

The authors

Round 2

Reviewer 2 Report

The authors answered some the questions but still some questions remained unanswered.

The contributions of the current work are not clear! There are many algorithms for heartbeat and respiration rate estimation from radar. What does the current paper exactly provide in addition to them? 

How does the method performs in movement artifacts is still a question.

The method is not even compared with the state-of-the art. What is the filtering technique the method is compared with?

Some figures are still very confusing. What is Figure 12 exactly? where is the noisy signal and where is the recovered signal? Why don't the authors plot the top noisy signal on the recovered signal so that the differences become clear?

Moreover, what is the noise source shown in the figure, given that there is not movement, where does the ugly signal comes from?

Reviewer 3 Report

The addition of other HF Radar applications is good but a significant one still remains. Therefore in line 33 please add to the text immediately prior to "naval search and rescue operations" the words "oceanographic surface current velocity measurements" and add the following comprehensive reference:

Liu, Y., Weisberg, R. H., and Merz, C.R., 2014: Assessment of CODAR Seasonde and WERA HF Radars in Mapping Surface Currents on the West Florida Shelf. J. Atmos. Oceanic Technol., 31, 1363–1382, doi:10.1175/JTECH-D-13-00107.1.

Author Response

Dear reviewer,

Thanks so much for the comments, and we are sorry for missing the comprehensive reference. According to your suggestions, we have added the suggested reference as reference [6] on Page 2 in Section 1. Please find the description on the revised version for detail. Thanks again for your time and consideration.

Best

The authors

Round 3

Reviewer 2 Report

The provided answers to the previous comments are too vague and unclear. For example authors mention that: "the doppler radar cannot calculate
the distance information of the human being, leading to severely micro-motion interference from the environment."  and then later on they say "the micro-motion interference from the human body and the nearby environment are inevitable for both the Doppler radar and FMCW radar based vital sign detection system" and in their long reply to comment #1, its really hard to find the real answer. The advantage of using FMCW radar is therefore unclear!

The same problem continues with too long and irrelevant answers to other questions.

I would suggest that authors re-write introduction. Those 4 points in the introductions are not contributions.

Author Response

(The authors gave the same response as above.)
